# Protocol for the Development and Analysis of the Oxford and Reading Cognitive Comorbidity, Frailty and Ageing Research Database-Electronic Patient Records (ORCHARD-EPR)

Emily Boucher ⓘ ,[1] Aimee Jell,[2] Sudhir Singh,[3,4] Jim Davies,[5] Tanya Smith,[6,7] Adam Pill,[6] Kinga Varnai,[8] Kerrie Woods,[8] David Walliker,[8] Aubretia McColl,[9,10] Sasha Shepperd ⓘ ,[11] Sarah Pendlebury ⓘ [1,3,4,12]

For numbered affiliations see end of article.

**Correspondence to**
Dr Sarah Pendlebury;
sarah.pendlebury@ndcn.ox.ac.uk

## ABSTRACT

**Background** Hospital electronic patient records (EPRs) offer the opportunity to exploit large-scale routinely acquired data at relatively low cost and without selection. EPRs provide considerably richer data, and in real-time, than retrospective administrative data sets in which clinical complexity is often poorly captured. With population ageing, a wide range of hospital specialties now manage older people with multimorbidity, frailty and associated poor outcomes. We, therefore, set-up the Oxford and Reading Cognitive Comorbidity, Frailty and Ageing Research Database-Electronic Patient Records (ORCHARD-EPR) to facilitate clinically meaningful research in older hospital patients, including algorithm development, and to aid medical decision-making, implementation of guidelines, and inform policy.

**Methods and analysis** ORCHARD-EPR uses routinely acquired individual patient data on all patients aged ≥65 years with unplanned admission or Same Day Emergency Care unit attendance at four acute general hospitals serving a population of >800 000 (Oxfordshire, UK) with planned extension to the neighbouring Berkshire regional hospitals (>1 000 000). Data fields include diagnosis, comorbidities, nursing risk assessments, frailty, observations, illness acuity, laboratory tests and brain scan images. Importantly, ORCHARD-EPR contains the results from mandatory hospital-wide cognitive screening (≥70 years) comprising the 10-point Abbreviated-Mental-Test and dementia and delirium diagnosis (Confusion Assessment Method—CAM). Outcomes include length of stay, delayed transfers of care, discharge destination, readmissions and death. The rich multimodal data are further enhanced by linkage to secondary care electronic mental health records. Selection of appropriate subgroups or linkage to existing cohorts allows disease-specific studies. Over 200 000 patient episodes are included to date with data collection ongoing of which 129 248 are admissions with a length of stay ≥1 day in 64 641 unique patients.

**Ethics and dissemination** ORCHARD-EPR is approved by the South Central Oxford C Research Ethics Committee (ref: 23/SC/0258). Results will be widely disseminated through peer-reviewed publications and presentations at conferences, and regional meetings to improve hospital data quality and clinical services.

## STRENGTHS AND LIMITATIONS OF THIS STUDY

⇒ ORCHARD-EPR overcomes the limitations of prospective cohorts, which are subjected to resource constraints, recruitment challenges and selection bias (eg, from the requirement for research consent) through providing multimodal rich clinical data at scale.

⇒ ORCHARD-EPR has complete coverage of the target population without selection and electronic follow-up, including through linkage to mental health records and brain imaging (CT, MRI, other) not routinely available in existing administrative data sets.

⇒ ORCHARD-EPR exploits the successful regional implementation of routine on-admission cognitive screening, which is often poorly captured adding further value to the rich phenotypic data collected as part of routine clinical care.

⇒ ORCHARD-EPR is a two-centre UK study and may not be generalisable to other populations, but the combined Oxfordshire and Berkshire populations are representative of England as a whole.

⇒ Free text entries including medical clerking and allied health assessments are currently unavailable for analysis, although information from these entries could eventually be extracted using artificial intelligence tools.

## INTRODUCTION

The advent of hospital electronic patient records (EPRs) offers the opportunity to exploit large-scale routinely acquired clinical data to improve patient care. Such data offer several advantages. First, they can be obtained at scale without additional burden to patients or staff and at relatively low cost.

Second, the use of EPR data (with the appropriate governance and ethical approvals) allows the entire hospital population to be studied without the unavoidable selection bias incurred in requiring individual consent for research assessments.[1] This is particularly advantageous in studies on older, frail and cognitively impaired patients in whom capacity to consent to research may be lacking, or ability to tolerate research assessments may be limited.[2 3] Third, obtaining data from the EPRs provides a considerably richer source of data than is currently available from hospital administrative data sets that are comprised largely of ICD-10-coded diagnoses, which have limitations especially in capturing complexity including frailty syndromes.[2 4 5] Fourth, EPRs provide real-time information allowing implementation of algorithms to identify those at high risk (eg, of delirium).[6]

The case-mix of general hospitals is changing across high-income countries in line with population ageing with implications for a wide range of hospital specialties.[7–9] Older people aged ≥65 years occupy the majority of hospital bed-days and many have complex conditions.[10] Up to a half have a cognitive disorder (eg, delirium, dementia) and around 40% are physically frail with an increased vulnerability to stressors such as a sudden decline in health.[2 11–16] Both cognitive and physical frailty are associated with a broad range of adverse health and social outcomes[2 11 13 15 16] and improving acute hospital care for older patients is a priority as reflected in policy documents and guidelines internationally.[17–22] Therefore, hospital data systems need to be established to identify frail patients across the hospital and across specialties, discriminate cognitive from physical frailty and ideally, the frailty domains affected given the implications for care.[2 11 17–22] In addition, collection of detailed clinical data including on diagnosis, comorbidities and illness acuity will aid understanding of the prognostic value of frailty across a range of acute hospital specialties and settings, improve risk stratification, inform secondary and tertiary prevention measures and provide data for research.[2 11 20]

In this protocol, we describe the methods underpinning the Oxford and Reading Cognitive Comorbidity, Frailty and Ageing Research Database—Electronic Patient Records (ORCHARD-EPR), which uses routinely acquired rich multimodal electronic individual patient data including routine cognitive/delirium screening, nursing risk assessments (falls risk, nutrition, pressure sore risk), observations, laboratory tests and brain imaging. ORCHARD-EPR will facilitate healthcare innovation and medical decision-making including the development/validation of risk prediction algorithms including for delirium and future dementia. ORCHARD-EPR will also enable key evidence gaps to be addressed around cognitive and physical frailty in the hospital setting as summarised in box 1 overcoming the limitations in existing studies from small selected samples (prospective studies) or reliance on hospital diagnostic ICD-10 coding (large retrospective studies).[4 5]

---

> **Box 1  Knowledge gaps to be addressed using ORCHARD-EPR**
>
> **Cognitive frailty**
> Prevalence of all-cause and subtypes of cognitive frailty in the acute hospital population overall and by age, sex and specialty.
> Accuracy of ICD-10 coding for cognitive diagnoses.
> Outcomes of all-cause cognitive frailty and by subtype (adjusted for confounding).
>
> **Overall frailty**
> Prevalence in the acute hospital population overall and by age, sex and specialty including same day emergency care (SDEC) as measured by different frailty tools (eg, Clinical Frailty Scale, Hospital Frailty Risk Score, Dr Foster score, frailty aggregate score).
> Prevalence of frailty markers identified from nursing assessments (eg, at risk of falls, pressure sores and malnutrition).
> Time trends in frailty prevalence in in-patient vs SDEC populations.
> Outcomes of frailty (adjusted for comorbidity burden, illness severity, comorbid dementia, care home residence).
> Prevalence and prognosis in key subgroups, that is, care home residents, comorbid dementia/delirium, severe illness.
>
> **Healthcare innovation**
> Development of predictive algorithms for delirium, and for dementia on follow-up.
> Use of routinely acquired brain imaging for the development of AI tools to quantify 'brain frailty' including atrophy and small-vessel disease and old stroke lesions to aid dementia subtyping and risk stratification.
> Real-time tracking of frail individuals through the hospital system to inform service delivery and design.
> Development of a real-time, 'streamlined Comprehensive Geriatric Assessment' to identify care needs using routinely acquired electronic patient record data.

## Aims and objectives

The primary aim of ORCHARD-EPR is to use individual patient EPRs to study the prevalence, complications and outcomes of cognitive and physical frailty in a large unselected, consecutive cohort of adults with unplanned hospital admission or emergency ambulatory care assessment.

The main objectives are to:

1. Estimate the hospital-wide prevalence of cognitive frailty (ie, delirium, dementia, delirium superimposed on dementia, low cognitive test score) by age, sex and specialty.
2. Estimate the prevalence of physical/global frailty using a variety of measures, including global scores (Clinical Frailty Scale–CFS),[23] and administrative diagnostic coding (ICD-10) data (Hospital Frailty Risk Score (HFRS),[16] coded frailty syndromes)[24] by age, sex and specialty.
3. Determine the associations of cognitive and physical frailty with outcomes (death, length of stay, readmission, increased care needs at discharge) adjusted for demographics, deprivation, multimorbidity, illness severity and other putative effect modifiers.[2 11]
4. Develop and integrate within the EPR a pragmatic tool (EPR-Comprehensive Geriatric Assessment (eCGA)) to identify frailty domains using routinely acquired

clinical EPR hospital data to target frailty interventions in real time.

5. Provide a rich research resource for further studies, including disease-specific studies (eg, on stroke), time trends in frailty prevalence and outcomes, investigation of risk factors for cognitive frailty (eg, neuroimaging markers and vascular disease), the development and validation of diagnostic and risk prediction tools (eg, for delirium and for dementia on follow-up) and health economics studies.

## METHODS AND ANALYSIS
### Study design and setting
The ORCHARD-EPR database was established by collecting pseudo-anonymised routinely acquired data extracted from the Oxford University Hospitals National Health Service (NHS) Foundation Trust (OUHFT) Cerner Millenium EPR (2015 onwards). Adult patients with unplanned admission or same day emergency care (SDEC) are included. The OUHFT is a large teaching and research trust consisting of four hospitals (John Radcliffe Hospital, Nuffield Orthopaedic Centre and Churchill Hospital in Oxford, and the Horton General Hospital in Banbury) providing all acute hospital care in Oxfordshire, UK with a catchment area of >800 000 people. There are short-stay nursing home beds for those requiring further assessment prior to discharge back to the community, and some specialty services to neighbouring counties.[25] The Oxfordshire population is older than the England average, all levels of deprivation are represented but with a lower average deprivation, and the urban/rural mix is in line with England as a whole.[26] ORCHARD-EPR is approved by the South Central Oxford C Research Ethics Committee (REC reference: 23/SC/0258). From 2024 onwards, we plan to extend ORCHARD-EPR to include the Royal Berkshire NHS Foundation Trust, with a larger catchment population (~1 million), subject to the necessary agreements. The combined Oxfordshire and Berkshire population is well matched to the overall English population demographic characteristics.

### Regulatory approvals
Extensive work (by SP, 2017-current) to meet regulatory requirements and overcome logistical barriers was required to set up ORCHARD-EPR. This has included:
► Initial OUHFT audit approval and registration to obtain the EPR data from the OUHFT Information team (AJ).
► Assigning of a unique study number (ORCHARD-EPR studyID) generated using an algorithm by the Information team (AJ) to enable deidentification of ORCHARD-EPR data.
► Extensive iterative checking of ORCHARD-EPR data accuracy in collaboration with the OUHFT Information team (SP, AJ).

► Working with the OUHFT clinical coding team (SP) to improve ICD-10 coding for cognitive frailty including delirium.[4]
► Data Protection Impact Assessment and Caldicott guardian sign-off for use of OUHFT EPR and brain imaging data and transfer of deidentified data to the University of Oxford servers.
► Ethics approval to use ORCHARD-EPR for research purposes with opt-out publicised on the OUHFT website and via posters.
► Approval from the Clinical Records Interactive Search (CRIS) oversight group for use of Oxford Health NHS Foundation Trust CRIS, Powered by Akrivia Health, mental health data in linkage to ORCHARD-EPR OUHFT data.[27]
► Data transfer agreement between Oxford Health NHS Foundation Trust and OUHFT for transfer of pseudonymised mental health CRIS data.
► Collaboration agreement between University of Oxford and OUHFT to cover use of OUHFT data in research that might generate intellectual property.
► Development of a pipeline to automatically deidentify, link to study ID and transfer brain scan images from the OUHFT Picture Archiving and Communication System to the University of Oxford servers (JD).
► Currently, the ethics approval for this study does not allow access to data to researchers outside the University of Oxford/OUHFT but widening access to researchers from other institutions is currently being explored.

### Participants and eligibility
The EPR was introduced to OUHFT in 2015, but complete coverage of all clinical areas and for all functionalities was not achieved until 2017. ORCHARD-EPR contains data on eligible patients from January 2015 with data added every 6 months and includes data covering the period of the COVID-19 pandemic and beyond.

### Inclusion
OUHFT in-patients and acute ambulatory care (SDEC) patients ≥65 years as well as patients <65 years with a completed cognitive screen (see later), and adult out patients with conditions increasing the risk of cognitive or physical frailty in which data have been collected as part of OUHFT-approved audits (see online supplemental file 1). Individual consent is not obtained for inclusion of pseudonymised data in the database although patients may opt out as stated earlier.

### Exclusion
Adult patients <65 years without a completed cognitive screen or test or otherwise at risk of cognitive or physical frailty.

### Study size
Over 200 000 unplanned hospital admission episodes and ambulatory care assessments are included in ORCHARD-EPR (2015-to date, with data collection

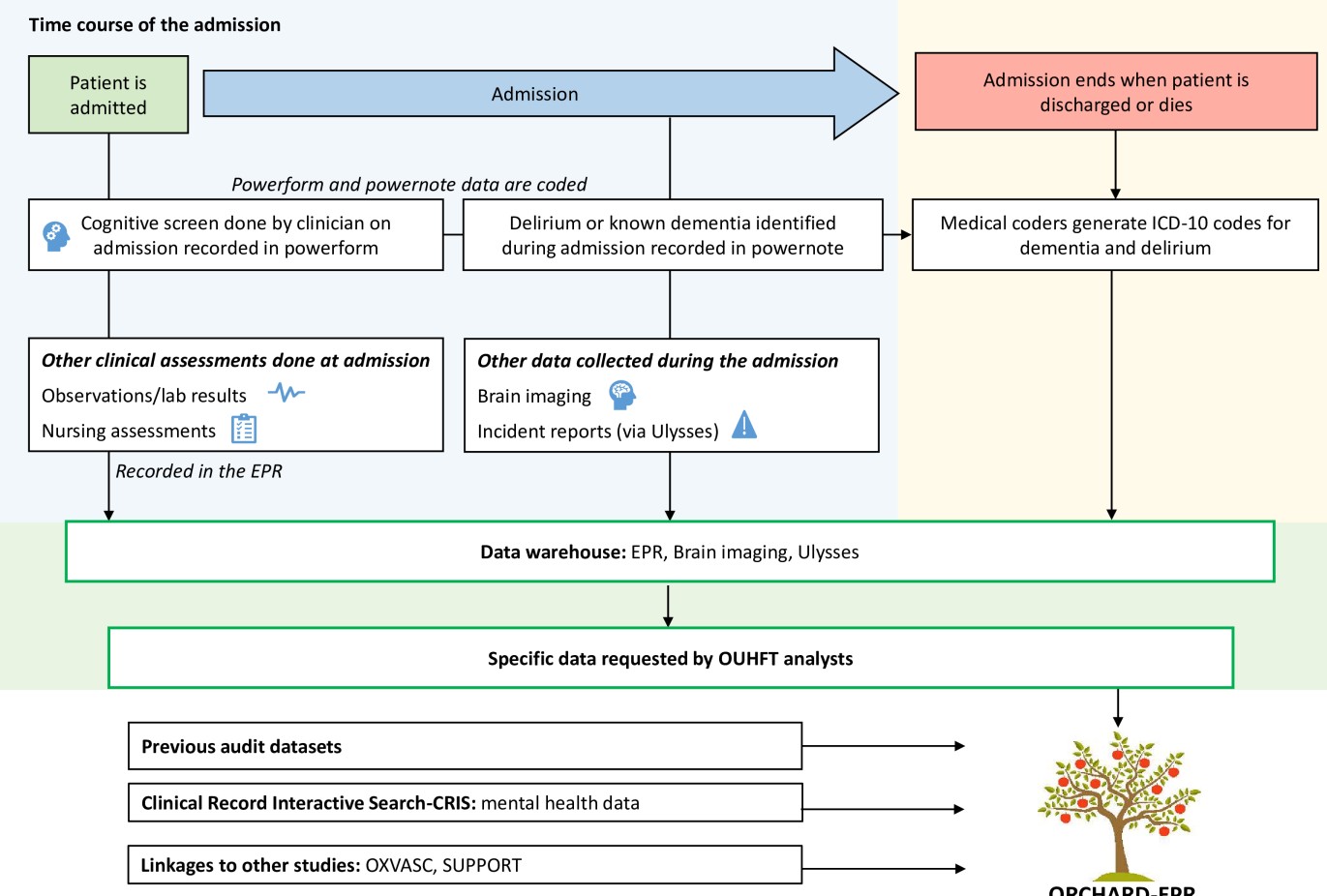

**Figure 1** ORCHARD-EPR—origin of data and creation of the database showing inclusion of clinical EPR data, observations and laboratory tests, brain imaging, secondary mental health diagnoses and relevant audits. Additionally, shows how cognitive data are ascertained in ORCHARD-EPR using the cognitive screening results together with administrative diagnostic (ICD-10) coding. CRIS, Clinical Records Interactive Search; ORCHARD-EPR, Oxford and Reading Cognitive Comorbidity, Frailty and Ageing Research Database-Electronic Patient Records; OUHFT, Oxford University Hospitals NHS Foundation Trust; OXVASC, Oxford Vascular Study; SUPPORT, Prospective Cohort Study of Proactive Supportive Care for Hospitalised Adults with Life-limiting Conditions: Evaluation of Patient Outcomes and Hospital Costs.

ongoing). Of these, 129 248 are inpatient admissions with a length of stay ≥1 day in 64 641 unique patients.

### Study procedures

All ORCHARD-EPR data are obtained from assessments completed by medical, nursing and allied health professional or administrative staff as part of routine patient care. Data are obtained from the EPR via the OUHFT information team or manually (>100 data fields in total, figure 1). The database contains data from EPR power-forms (ie, electronic proformas from which structured data can be extracted) but not powernotes (ie, free-text entries by clinicians, see online supplemental file 1). Individual patient records are assigned an ORCHARD-EPR study number by the Information Team and patient identifiers (eg, name, date of birth, hospital MRN, NHS number, etc) are removed before transfer to the database.

Table 1 summarises the data items collected in ORCHARD-EPR and whether they are available at multiple time points over an admission, and how these compare with the data available from UK-wide Hospital Episode Statistics-HES, which are similar to most international hospital administrative data sets. Key data fields extracted include demographics, postcodes, treating specialty, diagnoses, cognition, nutrition, falls and pressure sore risk, observations, illness severity, laboratory analyses and outcomes including length of stay. Importantly, ORCHARD-EPR contains extensive cognitive phenotyping data obtained through embedded hospital routine cognitive screening and linkage to mental health data as well as brain imaging.

### Cognitive screen

ORCHARD-EPR exploits previous extensive work (2010 onwards) led by SP to develop, validate and implement routine cognitive screening including for delirium for at-risk OUHFT patients (figure 2 and see below).[4 15 28–33] Since 2013, OUHFT has mandated that all patients with unplanned admission≥70 years of age or younger patients at high-risk (eg, alcohol excess, multiple sclerosis, stroke history) are administered the validated cognitive screen on admission.[4 15] This was initially delivered as part of

**Table 1** Data items available in ORCHARD-EPR, including at multiple time-points during admission, with comparison of availability in UK Hospital Episode Statistics (HES) administrative data predominantly composed of ICD-10 diagnostic codes

| Category | ORCHARD-EPR variable | Available at multiple time-points in ORCHARD-EPR | Availability in administrative datasets (eg, HES) |
|---|---|---|---|
| Demographic | Age, gender, ethnicity, postcode, deprivation | - | ✓ |
| Admission | Presenting complaint‡ | – | x |
|  | Method (ie, GP, ED), source | – | ✓ |
| Vitals | HR, BP, RR, SpO$_2$, tympanic temp, AVPU | ✓ | x |
| Illness severity | NEWS, SIRS | ✓ | x |
| Cognitive screen | AMTS, MMSE, MoCA, ACE | ✓ | x |
|  | Reason for cognitive untestability | ✓ | x |
|  | Delirium diagnosis | ✓ | Limited* |
|  | Delirium on admission (prevalent) and arising de novo during admission (incident)‡ | ✓ | x |
|  | Pre-admission dementia diagnosis | – | Limited* |
| Multimorbidity | Charlson, Elixhauser indices | – | ✓ |
| Frailty | HFRS | – | ✓ |
|  | Clinician's impression of frailty‡ | – | x |
|  | Dr Foster Global Frailty Score | – | ✓ |
|  | Frailty Markers, including vision and hearing | – | Limited |
|  | Long-term catheter (medical/nursing notes)‡ | ✓ | x |
|  | Catheter insertion during admission (medical/nursing notes)‡ | ✓ | x |
| Functional status | Nursing risk assessments | ✓ | x |
| Nutrition | Malnutrition universal screening tool (MUST) | ✓ | x |
| Pressure sore risk | Braden scale (pressure sore risk), including sensory perception, moisture, activity, mobility, nutrition, friction and shear | ✓ | x |
| Falls risk factors (based on NICE) | Orthostatic hypotension, appropriate footwear, safe walking unaided/with an aid, urinary incontinence or urgency, vision problems, fall history, fear of falling, environment (hazards, lighting, call bell), confusion/agitation/cognitive impairment | ✓ | x |
| Mental capacity | Documented mental capacity assessment yes/no, decision assessed,‡ outcome of decision‡ | ✓ | x |
| Course in hospital | Ward | ✓ | x |
|  | Specialty (ie, treatment function, main specialty) | – | ✓ |
|  | Procedures, including radiologic imaging | ✓ | ✓ |
|  | Incidents in hospital | ✓ | x |
| Diagnoses | ICD-10 codes | – | ✓ |
|  | Clinician working diagnosis and final diagnosis‡ | – | x |
| Disposition | Residence‡ | – | x |
|  | Care needs/care package‡ | – | x |
|  | Increased care needs at discharge: new care home, rehabilitation bed, intermediate care, increased care package‡ | ✓ | x |
|  | LOS in hospital, critical care, palliative care, rehabilitation, | – | ✓ |

Continued

**Table 1** Continued

| Category | ORCHARD-EPR variable | Available at multiple time-points in ORCHARD-EPR | Availability in administrative datasets (eg, HES) |
|---|---|---|---|
| | Delayed transfer of care, reasons for delayed transfer | ✓ | x |
| | Discharge destination | – | ✓ |
| | Death (via National Office of Statistics) | – | ✓ |
| **Labs** | Haemoglobin, white cell, neutrophil and lymphocyte counts, albumin, C reactive protein, creatinine, eGFR, glucose, HbA1c, lactate, pH, sodium, thyroid stimulating hormone, urea | ✓ | x |
| **Brain imaging** | Brain imaging reports, raw brain scan images | ✓ | x |
| **Mortality** | During admission and on follow-up | – | ✓ |
| **Long-term cognitive/mental health outcomes** | Dementia and mild cognitive impairment from mental health services assessment | ✓ | Limited† |
| | Cognitive test scores | ✓ | x |
| | Depression and mental illness | ✓ | Limited† |

CRIS uses artificial intelligence to extract data from free text.
*Only available via clinical coding that is known to be insensitive.
†Only available for patients readmitted to hospital in whom a corresponding ICD-10 diagnosis was recorded by the coders.
‡Only available in manual audits.
ACE, Addenbrooke's Cognitive Examination; AMTS, Abbreviated Mental Test Score; AVPU, Alert, Voice, Pain, Unresponsive Scale; CRIS, Clinical Record Interactive Search; eGFR, Estimated glomerular filtration rate; HbA1c, Hemoglobin A1C; LOS, Length of stay; MMSE, mini-mental-state examination; MoCA, Montreal Cognitive Assessment; MUST, Malnutrition Universal Screening Tool; NEWS, National Early Warning Score; NICE, National Institute for Health and Care Excellence; ORCHARD-EPR, Oxford and Reading Cognitive Comorbidity, Frailty and Ageing Research Database-Electronic Patient Records; SIRS, Systemic Inflammatory Response Syndrome.

a paper clerking proforma (2013–2015) designed by SP and SSi but since 2015, this has been integrated as a powerform into the EPR (28 466/40 439 (70%) eligible patients staying ≥1 day screened for period 2017–2019).[34] Ongoing performance feedback and governance reviews are in place to ensure that screening rates are maintained.

The OUHFT cognitive screening EPR powerform (online supplemental figure 1) includes:

▶ The 10-point Abbreviated Mental Test Score,[35] a brief screen for cognitive problems or drop-down list for reason for lack of completion (eg, patient too unwell, aphasia).

▶ Recording of formal dementia diagnosis (recorded as yes/no/uncertain).

▶ Recording of delirium diagnosis (recorded as yes/no/uncertain) informed by the CAM.[36] Since screening is mandated on admission, prevalent (on-admission) delirium is more likely to be recorded via the powerform than incident delirium (arising during admission).[4] Incident delirium may be recorded via launching another cognitive screening powerform or may be documented in a free text powernote form, which is then assigned an ICD-10 delirium code by coding staff (figure 1).

Training in delirium diagnosis, using the CAM, and according to the Diagnostic and Statistical Manual of Mental Disorders, (DSM-IV, now DSM-5) criteria, is provided to junior medical staff during foundation and middle grade training and this together with trust-wide delirium education and awareness raising was implemented as part of a multicomponent intervention:[4] staff education/training including of the coding team, implementation of mandatory cognitive screening via the clerking proforma and then the EPR powerform, and introduction of regular performance feedback on rates of screening at hospital governance meetings, figure 2. Other cognitive screening tests built as powerforms in EPR including the mini-mental-state examination (MMSE),[37] Montreal Cognitive Assessment (MoCA),[38] Addenbrooke's Cognitive Examination (ACE-III)[39] and Informant Questionnaire for Cognitive Decline in the Elderly (IQCODE)[40] are also included in ORCHARD-EPR where available.

## Nursing risk assessments

ORCHARD-EPR includes data from relevant nursing risk assessments, including the Malnutrition Universal Screening Tool,[41] Braden Scale for pressure sore risk[42] and a multifactorial falls risk assessment based on National Institute for Health and Care Excellence guidance.[43] These include data on functional status, vision and continence.

## Derived variables

Laboratory test results and physiologic measures, including heart rate, blood pressure and temperature,

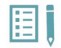

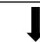

Development/validation/pilot of cognitive screen, 2010-2012
- Feasibility
- MMSE vs AMTS vs MoCA vs IQCODE
- CAM for delirium

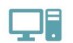

Implementation (paper), 2013
- Paper clerking proforma rolled-out for AGM
- Cognitive screen integral part of paper clerking proforma

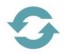

Implementation (electronic), 2015
- Roll-out of EPR
- Cognitive screen built into EPR as a powerform
- Flagged as a mandatory task on admission:
  - All patients aged ≥70 years or younger at-risk
  - Unplanned admissions
  - All specialties

Performance monitoring, 2013 onwards
- Screening figures reported nationally, 2012-2019
- Feedback at clinical governance meetings
- Monthly reporting of screening compliance
- Ongoing QIP projects

**Figure 2** Development, validation and implementation of the cognitive screen. In two audit cycles undertaken in acute general medicine (AGM) in 2010 and 2012, the CAM was piloted in conjunction with first the MMSE and then the AMTS. The AMTS was then compared with the MoCA and the IQCODE to determine its sensitivity and specificity for mild and moderate/severe cognitive impairment.[28 31] The cognitive screen comprising the AMTS+CAM was subsequently implemented in AGM for all those aged ≥70 years via a paper clerking proforma, and from 2015, hospital-wide via an EPR powerform. AMTS, Abbreviated Mental Test Score; CAM, Confusion Assessment Method; EPR, electronic patient record; IQCODE, Informant Questionnaire for Cognitive Decline in the Elderly; QIP, Quality Improvement Project; MMSE, mini-mental-state examination; MoCA, Montreal Cognitive Assessment.

from routine patient observations on admission are included allowing for the calculation of derived variables. These data are used to calculate measures of illness severity, including the:
► National Early Warning Score.[44]
► Systemic Inflammatory Response Syndrome criteria[45] (white cell count, blood pressure, respiratory rate, temperature).
► Acute Physiology and Chronic Health Evaluation II (APACHE II).[46 47]
► Quick Sequential Organ Failure Assessment (qSOFA) score.[48]

Since such data are available throughout admission, they can be used to generate dynamic data on intra-admission trajectories. ICD-10 codes are allocated by OUHFT coding administrators and are used to calculate the Charlson comorbidity index with updated weights,[49] and the HFRS.[16] Postcodes are collected as part of admission demographics and to allow calculation of the English Index of Multiple Deprivation (2019) to obtain percentiles of deprivation.[50] Postcodes are also used to cross-check against social services lists of care homes to assign care home residence pre-admission.[51]

## Data linkage

ORCHARD-EPR linkage to the Oxford Vascular Study (OXVASC) transient ischemic attack (TIA)/stroke cohorts has already provided long-term follow-up data on hospital admission after TIA and stroke to inform delirium, infection and future dementia risk associations.[29 30 52]

ORCHARD-EPR is also linked to:
► local mental health trust data (Oxford-Health-CRIS);
► the OUHFT incident reporting database (Ulysses);
► Criteria for Screening and Triaging to Appropriate aLternative care scores collected by the OUHFT Palliative care team.[53]
► routinely acquired CT and MRI brain imaging.

Oxford-Health CRIS contains deidentified, pseudonymised data on mental health diagnoses including dementia, mild cognitive impairment, cognitive test scores, depression and other psychiatric diagnoses including extracted from free text inpatient and outpatient clinical records using artificial intelligence methods.[27] Linkage to CRIS will allow reliable identification of dementia preadmission and on follow-up. Brain scan images (n>1300 linked CT-brain and MRI-brain scans to date) will allow incorporation of brain ageing/neuropathology markers. Linkage to the OUHFT incident reporting database (Ulysses) will provide data on adverse events in hospital, including falls (severity, associated harm), pressure ulcers and challenging patient behaviours. Linkage of ORCHARD-EPR to other OUHFT and University of Oxford-approved studies will provide data on acute hospital illness and cognitive and physical frailty, where participants have given consent for access to their medical records.

## Outcome measures

Data are collected for the following outcomes:[2 20]
► Mortality obtained from the Office of National Statistics, which registers all deaths in the UK.
► Length of stay calculated from the time of admission to the time of discharge or transfer.
► Delayed transfers in care where a patient is determined to be ready to go home by their clinician and multidisciplinary team, but remains in a hospital bed, for example, because their care package is not ready;
► Discharge destination to public or private care homes or other hospitals (eg, for rehabilitation);
► Readmissions identified from ORCHARD-EPR including within 30 days of discharge;
► Cognitive decline and dementia from ORCHARD-EPR readmissions and Oxford Health (CRIS) data.

## Inclusion of previous audits

Additional pseudonymised data from OUHFT-approved audits around frailty and outcomes are transferred to ORCHARD-EPR. Such audit data include comparisons of the OUHFT cognitive screen with additional objective cognitive testing (eg, MMSE, IQCODE)[28 31] and rates of delirium occurrence in acute general medicine (n=6

cycles to date) with delirium diagnosis made by a consultant physician according to the DSM-IV criteria.[4 15] These audit data will act as the reference standard in evaluating the accuracy of the ORCHARD-EPR routinely acquired real-world EPR data (see below).

## Data accuracy

ORCHARD-EPR accuracy is carefully verified using the following:

► Selection and identification of data fields for extraction is done by SP, a clinician working in acute and complex medicine who uses the EPR interface on a daily basis, and has led implementation of changes to EPR;

► Extracted data is checked for accuracy by SP in collaboration with the information team analysts (AJ, ie, for missing data, and sense-checking of extracted values);

► Extracted data are verified against prospective audit data (eg, for rates of delirium and diagnosed dementia)[15] and against data extracted independently by the Information team for other purposes (eg, monthly cognitive screening rates).

## Statistical analyses

Demographic and patient characteristics will be presented using descriptive statistics ($\chi^2$ for categorical data and t-test/Analysis of Variance–ANOVA for continuous data as appropriate). Prevalence of cognitive frailty syndromes (ie, any cognitive syndrome/deficit, dementia, delirium, delirium superimposed on dementia, low objective test score) will be calculated overall and stratified by age, sex and specialty with 95% CIs. To determine the association between frailty syndromes (independent variables) and the outcomes listed above (dependent variables), risk ratios will be modelled using logistic, Poisson and Cox regression. Effect modification and confounding by age, sex, deprivation, comorbidity, residence at the time of admission and illness severity will be evaluated. Statistical analysis methods will be updated over time, as new data are added and research questions identified.

## Adherence to reporting guidelines

All research will be reported in accordance with relevant guidelines including Strengthening The Reporting of OBservational Studies in Epidemiology[54] and Transparent Reporting of a multivariable prediction model for individual prognosis or diagnosis guidelines.[55]

## Patient and public involvement

ORCHARD-EPR has been informed since its inception by a patient and public involvement (PPI) group, set up in 2015 by SP, to help develop studies on cognitive and physical frailty. PPI members provided input into the ORCHARD-EPR proposal and objectives. The PPI group comprises patients with cognitive and physical frailty with lived experience of hospitalisation and their carers, a retired general practitioner, Alzheimer's society representatives and a PPI lead with experience in frailty and dementia studies. Discussions with the PPI group

highlighted that a key concern for patients and families is the need for better information about prognosis in relation to frailty and confusion in hospital. The PPI group has also facilitated several projects that have impacted the richness and depth of ORCHARD-EPR data including the system-wide multicomponent intervention to improve the recognition, recording and diagnostic coding of delirium and other cognitive frailty syndromes, implementation of cognitive screening in acutely unwell patients and development of the delirium susceptibility score, now implemented as an EPR digital algorithm to flag patients at risk in real time.[4 6 15]

## DISCUSSION

### Summary

ORCHARD-EPR is unique in terms of its size, inclusiveness and scope (>200 000 unselected unplanned patient episodes, 2015–2021) with extensive real-world multimodal routinely acquired EPR data largely unavailable in administrative data sets based on ICD-10 diagnostic coding (table 1). ORCHARD-EPR combines detailed phenotyping for cognitive and physical frailty and other clinical characteristics with physiologic and laboratory measures, illness severity, care home residence, brain scan images and outcomes including mental health diagnoses from linkage to mental health secondary care data. ORCHARD-EPR will enable large-scale studies on the prevalence and outcomes of cognitive and physical frailty including delirium, with robust adjustment for confounding thereby filling existing knowledge gaps. ORCHARD will also enable the development of risk prediction algorithms (eg, for dementia) and disease-specific studies through selection of appropriate subgroups or through linkage to existing cohorts.

### Cognitive frailty

Cognitive frailty has multiple direct implications for patient care, service planning and policy including capacity to consent for procedures, medication use, discharge planning, communication and advanced directives/proactive palliative care.[2 13 15 20] Despite its importance, however, reliable estimates in unselected cohorts of its prevalence (and of relevant subtypes, particularly delirium) including by specialty are scarce.[2] Existing prospective studies of highly phenotyped cohorts tend to have small sample sizes and restrictive eligibility criteria, resulting in exclusion of high risk groups such as those unable to give informed consent.[1–3] Retrospective administrative data sets (eg, UK HES data) are large and inclusive but are usually unreliable for ascertainment of cognitive frailty because of the insensitivity of ICD-10 diagnostic coding or reliance on non-specific free text terms (eg, 'disturbed behaviour').[2 4 5]

ORCHARD-EPR addresses the limitations of previous data sets through combining the detailed cognitive phenotyping data usually only available in small, labour-intensive prospective studies with the large sample size,

inclusivity and efficiency of retrospective administrative 'Big Data' studies. Specifically, the value of the electronic cognitive data in ORCHARD-EPR is greatly enhanced by several factors. First, ORCHARD-EPR exploits routine hospital-wide cognitive screening including for delirium embedded via an electronic powerform, implemented using a multicomponent intervention ensuring better recognition and documentation of cognitive frailty. Second, ORCHARD-EPR ascertains cognitive frailty using both the results of the EPR cognitive screen powerform and the ICD-10 diagnostic coding applied by the hospital coding team who have been trained by SP ensuring that free text powernote entries documenting cognitive frailty are translated into the appropriate ICD-10 code. Third, the accuracy of ORCHARD-EPR cognitive frailty data is assessed against reference standard data provided by prospective audit (using analogous methodology to that used in the development and validation of the HFRS).[15 16] Fourth, linkage to secondary care mental health data provides enhanced information on dementia and other mental health conditions enhancing the sensitivity of cognitive outcomes.

### Physical frailty/frailty domains

In comparison to cognitive frailty, established methods are available to identify (global) frailty in large unselected retrospective administrative datasets. The widely used HFRS uses a combination of a broad range of administrative diagnostic (ICD-10) codes known to be present in those with frailty but is dependent on coding accuracy, resulting in heterogeneity across studies.[2 16] In addition, the HFRS provides a single numeric measure of frailty (which can be stratified as mild/moderate/severe) rather than providing information on specific frailty domains and, therefore, cannot inform care in real time during admission. In contrast, the rich routinely acquired clinical data in ORCHARD-EPR could be used to derive an eCGA,[56] in which frailty is identified across a range of domains (cognition, continence, mobility, vision, nutrition, falls risk etc) without further burden to patients or staff. Such an eCGA could be implemented into EPR to identify domains of need in real time to better target frailty interventions in collaboration with patients and families.

### Frailty trajectories, time trends and prognosis

Reliable capture of cognitive frailty in ORCHARD-EPR will allow evaluation of the extent of overlap with physical frailty, individual frailty trajectories across multiple admissions, and time trends in frailty prevalence in which there are few studies. Hospital frailty prevalence is likely to be increasing given the ageing population, and the introduction of ambulatory care, SDEC and Hospital-at-Home/virtual ward services which tend to be accessed by younger, fitter people.[32] ORCHARD-EPR will also provide a better understanding of the prognostic value of cognitive and physical frailty and their interactions with the potential for improved risk stratification.[2 11] Inclusion of

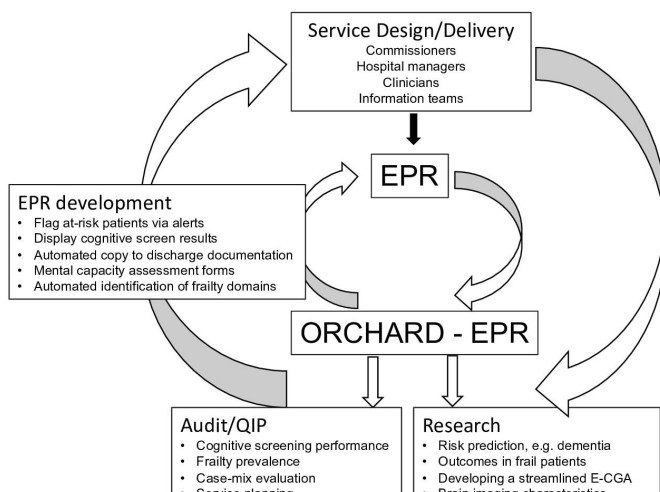

**Figure 3** Virtuous circle—the uses of ORCHARD-EPR. ORCHARD-EPR data can drive audit and research and the findings used to inform policy, commissioning and service planning/development as well as EPR developments to improve documentation and care. CGA, Comprehensive Geriatric Assessment; ORCHARD-EPR, Oxford and Reading Cognitive Comorbidity, Frailty and Ageing Research Database-Electronic Patient Records; QIP, Quality Improvement Projects.

more than 100 covariates allows robust adjustment for confounders and exploration of potential effect modifiers impacting outcomes such as care home residence and illness severity.[2] Linkage to other local hospital datasets (eg, incident reporting for in-patient falls, challenging behaviour) will provide reliable information on the impact of cognitive and physical frailty on healthcare safety and quality to inform care bundles and policy.

### Data accuracy and creating a virtuous circle

Reliable data are key to NHS management, service evaluation, planning and development, audit and research and the advent of hospital EPRs enables rich routinely acquired data to be exploited. However, healthcare staff entering data will tend to prioritise immediate patient care which may impact data accuracy.[57] ORCHARD-EPR data accuracy is checked through close working between SP, a clinician on the ground using the EPR user interface on a daily basis, and the hospital Information Team (AJ). This enables selection of data fields of interest and the necessary iterative work to be undertaken to ensure data fields are correctly identified and populated. Accuracy can be further checked against the reference standard of local prospective audit data.[15] ORCHARD-EPR aims to create a virtuous circle whereby audit and research informs EPR developments, improves data quality and ultimately improves patient care (figure 3). Further, since the EPR is a dynamic entity, ORCHARD-EPR can be used to drive future EPR developments to improve data capture as evidenced by the implementation of the EPR cognitive screen to improve recognition, documentation and administrative diagnostic coding of cognitive frailty. Delirium recognition has also been enhanced by the

development, validation and EPR implementation of the delirium susceptibility score algorithm to identify those at risk in real time.[6]

## Additional research questions

ORCHARD-EPR will enable researchers to explore a large number of additional research questions in the acute secondary care setting in a large, representative and inclusive population of older adults. Examples include outcomes in disease-specific studies (eg, COVID-19) and validation and recalibration of commonly used prognostic tools, (eg, the CURB-65 tool for mortality risk in pneumonia).[58] The availability of routinely acquired unselected brain imaging will allow development of artificial intelligence (AI) tools to extract markers of brain pathology (or "brain frailty") such as cerebral small vessel disease, brain atrophy and stroke lesions in representative older patient cohorts. The resultant imaging AI tools could be integrated into clinical imaging systems at source to provide quantitative data to aid neuroradiologists/clinicians without the need for human input. Inclusion of brain imaging data with linked clinical data will enhance the development of risk prediction tools (eg, for delirium and dementia) to improve care and selection for trials and understand mechanisms. ORCHARD-EPR will facilitate development of algorithms for in-hospital falls risk prediction/prevention and an eCGA to identify frail patients and their needs in real time without the need for additional assessments to target individualised interventions.[56] Finally, ORCHARD-EPR could act as an exemplar for the Subnational Secure Data Environments currently being setup across England to harness the power of NHS data for the benefit of patients.[59]

## Strengths and limitations

This paper has outlined the strengths of ORCHARD-EPR in detail. However, there are some limitations. Not all patients have cognitive screening because of service pressures, or lack of clinician engagement, and screening might be less likely in certain patient groups including the severely ill, resulting in non-random missingness and bias. However, screening rates have been high (~75% of those aged ≥75 years staying ≥72 hours) as documented by NHS England (2017–2019)[35] and ICD-10 coding for cognitive frailty has been substantially improved by a multicomponent intervention.[4] Second, the CFS[23] was only introduced during the COVID pandemic (2020 onwards) and is only now becoming routinely administered. Third, patients aged <70 years do not get routine cognitive screening and cognitive data are, therefore, limited to those deemed as at risk by clinicians. However, the age cut-off has now been lowered to 65 years and over (2024 onwards), so cognitive data will be routinely available on those aged 65–69 years going forward. Fourth, medications are not currently captured in ORCHARD-EPR, but there are plans to include these data in the future. Finally, free-text entries including the medical clerking and entries from allied health and specialist personnel

are currently unavailable, although AI methods could eventually be used to extract analysable data as is done for mental health (CRIS) data.[27]

## ETHICS AND DISSEMINATION

See 'regulatory approvals' heading above—ORCHARD-EPR is approved by the South Central Oxford C Research Ethics Committee (REC reference: 23/SC/0258). Results from this study will be disseminated through peer-reviewed publications, presentations at national and international conferences as well as regional meetings to improve hospital data quality and clinical services.

**Author affiliations**
[1]Wolfson Centre for Prevention of Stroke and Dementia, Nuffield Department of Clinical Neurosciences, University of Oxford, Oxford, UK
[2]Informatics Department, Oxford University Hospitals NHS Foundation Trust, Oxford, UK
[3]Department of Acute General (Internal) Medicine, Oxford University Hospitals NHS Foundation Trust, Oxford, UK
[4]Department of Geratology, Oxford University Hospitals NHS Foundation Trust, Oxford, UK
[5]Department of Computer Science, University of Oxford, Oxford, UK
[6]Research Informatics Team, Research and Development Department, Oxford Health NHS Foundation Trust, Oxford, UK
[7]NIHR Oxford Health Biomedical Research Centre, Oxford, UK
[8]Research and Development Clinical Informatics, Oxford University Hospitals NHS Foundation Trust, Oxford, UK
[9]Department of Acute Medicine, Royal Berkshire NHS Foundation Trust, Reading, UK
[10]Department of Elderly Care Medicine, Royal Berkshire NHS Hospital Foundation Trust, Reading, UK
[11]Nuffield Department of Population Health, University of Oxford, Oxford, UK
[12]NIHR Oxford Biomedical Research Centre, Oxford University Hospitals NHS Foundation Trust, Oxford, UK

**Contributors** EB assembled and cleaned the ORCHARD-EPR data and drafted the manuscript. AJ identified data variables within the OUHFT data warehouse and extracted and pseudonymised individual patient EPR data prior to transfer to ORCHARD-EPR. SSi provided OUHFT management support to embedding OUHFT-wide cognitive screening and ensuring compliance and in the set-up of ORCHARD-EPR. JD located, de-identified, pseudonymised with studyID and securely transferred brain images from the hospital PACS system to the University of Oxford servers. TS (Head of Research Informatics) and AP (Informatics Analyst) led the linkage to Oxford Health mental health data and secure transfer of pseudonymised mental health data using data systems from the NIHR Oxford Biomedical Research Centre, KV, KW, DW provided support with obtaining OUHFT EPR data, AM provided support with the planned Royal Berkshire Hospitals data linkage. SSh provided critical input to the manuscript. SP developed and implemented OUHFT-wide routine cognitive screening, led the multicomponent intervention to improve compliance and ICD-10 coding of delirium and other cognitive frailty, conceived and set-up ORCHARD-EPR, obtained funding, worked with AJ to ensure data accuracy and drafted the manuscript. All authors read and approved the final manuscript.

**Funding** EB is supported by the Rhodes Trust. SP is supported by the NIHR Oxford Biomedical Research Centre. The assembly of ORCHARD-EPR and its use for development of dementia risk prediction algorithms in older hospital patients is funded as part of a 5-year NIHR Invention for Innovation Programme grant to SP: NIHR204290, 2023-2028. The research was supported by the National Institute for Health Research (NIHR) Oxford Biomedical Research Centre (BRC) and funded by the NIHR i4i [NIHR204290] and John Fell fund [0012734]. The views expressed are those of the authors and not necessarily those of the NHS, the NIHR or the Department of Health.

**Competing interests** None declared.

**Patient and public involvement** Patients and/or the public were involved in the design, conduct, reporting, or dissemination plans of this research. Refer to the Methods section for further details.

**Patient consent for publication** Not applicable.

**Provenance and peer review** Not commissioned; externally peer-reviewed.

**Data availability statement** Currently, the ethics approval for this study does not allow access to data to researchers outside the University of Oxford/OUHFT but widening access to researchers from other institutions is currently being explored.

**ORCID iDs**
Emily Boucher http://orcid.org/0000-0002-9854-3462
Sasha Shepperd http://orcid.org/0000-0001-6384-8322
Sarah Pendlebury http://orcid.org/0000-0003-3603-8388

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
