## [Reviewer comments · BMJ Open]

ARTICLE DETAILS

TITLE (PROVISIONAL)	Protocol for the Development and Analysis of the Oxford and Reading Cognitive Comorbidity, Frailty and Ageing Research Database - Electronic Patient Records (ORCHARD-EPR)
AUTHORS	Boucher, Emily; Jell, Aimee; Singh, Sudhir; Davies, Jim; Smith, Tanya; Pill, Adam; Varnai, Kinga; Woods, Kerrie; Walliker, David; McColl, Aubretia; Shepperd, S; Pendlebury, Sarah

VERSION 1 – REVIEW

REVIEWER	Fogg, Carole University of Southampton, School of Health Sciences
REVIEW RETURNED	12-Mar-2024

GENERAL COMMENTS	Comments to the author: This novel research database looks like a fantastic opportunity to further research into the epidemiology of cognitive and physical frailty and to inform the development of more appropriate in-hospital care for older people. I had just a few minor queries: 1. As medications/polypharmacy are often a key part of the CGA and other assessments for older people, are there plans to include this data in future data capture? Perhaps this can go into the limitations as not currently captured. (Apologies if it was there and I missed it!)2. Could you please clarify further who amongst patients aged <65 would be considered “at risk” – examples of clinics are given but this is not exhaustive – there must be a specific list of criteria to be able to filter them into the database – this could be put in the supplementary material?3. Study procedures: What are “EPR powerforms”? “Pownote entries”? Presumably something related to the Cerner system, but please give a brief description of what this means. Is the intention to make free-text entries available also?4. If date of birth is removed from the EPR information at the anonymisation stage, how is patient age calculated / represented in the database?5. The database starts from 65+ but the cognitive screening information starts aged 70+ - are there plans to expand the coverage down to 65+? Otherwise will be difficult to use patients aged 65-69 in many of the analyses which require cognitive information as a covariate? Or will alternative methods/data be used for these patients? Does this need to go into the limitations also?
--

	6. Outcome measures: Could this please be re-organised with the definition/detail of each outcome next to the bullet point for easier reading? 7. The sentence “All participants are followed until death or the end of the specified study period” seems more appropriate for specific cohort studies which may be run using the data from the ORCHARD-EPR rather than a protocol for a research database, as presumably this is an open-ended database? Typos: Exclusion: “at-risk” change to “at risk” Study size – should be “in 64,641 unique patients”? (remove ‘over’)
--	--

REVIEWER	Best, Kate University of Leeds Faculty of Medicine and Health, Academic Unit of Ageing and Stroke Research
REVIEW RETURNED	25-Mar-2024

GENERAL COMMENTS	The protocol describes an very interesting data source that includes variables not available in other secondary care data sources such as HES. I have a few comments to improve clarity: Please could you add a basic description of powerform and powernote. Estimation of prevalence is listed as a key objective- could you clarify that this refers to hospital prevalence. ORCHARD-EPR is described as a data resource for researchers- are there plans to make the data accessible to those outside of the research team? Are there plans to record ethnicity?
---

VERSION 1 – AUTHOR RESPONSE

Reviewer: 1

Dr. Carole Fogg, University of Southampton

Comments to the Author:

This novel research database looks like a fantastic opportunity to further research into the epidemiology of cognitive and physical frailty and to inform the development of more appropriate in-hospital care for older people.

Thank you for the positive response about the research database and its potential utility.

I had just a few minor queries:

1. As medications/polypharmacy are often a key part of the CGA and other assessments for

older people, are there plans to include this data in future data capture? Perhaps this can go into the limitations as not currently captured. (Apologies if it was there and I missed it!)

Response: We appreciate the reviewer's suggestion and agree with the importance of medications/polypharmacy in old age medicine. We have not yet been able to extract these data but we are currently in discussions with the OUHFT Information analysts about extracting these data. We have added to the following text to the Discussion, Limitations section:

Discussion, Page 17, second paragraph: *"Fourth, medications are not currently captured in ORCHARD-EPR, but there are plans to include these data in the future."*

2. Could you please clarify further who amongst patients aged <65 would be considered "at risk" – examples of clinics are given but this is not exhaustive – there must be a specific list of criteria to be able to filter them into the database – this could be put in the supplementary material?

Response: For patients aged ≥ 65 years with unplanned admission or acute ambulatory care (same day emergency care-SDEC) assessment, and those aged <65 years with a completed cognitive screen, data are automatically filtered in to the database by the Information Analysts. However, the database also includes data from relevant OUHFT-approved audits on adult out-patients at-risk of cognitive and physical frailty defined as attending a vascular specialty clinic (eg TIA/minor stroke, peripheral vascular disease), memory clinics or geriatric medicine clinics where multimorbidity (the presence of two or more long-term conditions) is common. These audit data are collected manually since the EPR system does not currently record out-patient assessment information in a structured format. The de-identified data are then transferred to the database.

We have revised the text in the Manuscript as follows:

Methods/Participants and eligibility, page 7: "Inclusion: OUHFT in-patients and acute ambulatory care (SDEC) patients >65 years, as well as patients <65 years with a completed cognitive screen (see later), and adult out-patients with conditions increasing the risk of cognitive or physical frailty in which data have been collected as part of OUHFT-approved audits (see **Supplement**)."

We have added text to the Supplement as suggested by the reviewer to explain the inclusion criteria in more detail as follows:

"Supplemental Methods- ORCHARD-EPR Inclusion Criteria

Patients with unplanned admission or Same Day Emergency Care-SDEC attendance

All OUHFT in-patients with unplanned admission and acute ambulatory care (Same Day Emergency Care-SDEC) patients aged ≥ 65 years are included in the database with the relevant datafields extracted from the EPR by the OUHFT Information analysts. Similar data are automatically extracted for in-patients and SDEC patients aged <65 years with a completed cognitive screen indicating clinician concern about possible cognitive frailty (eg because of Parkinson's disease, stroke, multiple sclerosis, alcohol excess etc).

Pseudonymised data from previous relevant OUHFT-approved audits conducted prior to the implementation of EPR are also included, for example to establish the prevalence of delirium and cognitive impairment using different cognitive tests in consecutive admissions to acute general medicine.^{1,2}

Out-patients

In addition, relevant OUHFT-approved audit data may be included on out-patients (including patients aged <65 years) deemed to be at-risk of cognitive or physical frailty as indicated by referral to vascular specialty clinics (eg TIA/minor stroke, peripheral vascular disease), memory clinics or geriatric medicine clinics where multimorbidity (the presence of two or more long-term conditions) is common. Manual extraction of audit data by the usual care team is necessary since at present, EPR recording of out-patient assessments is generally in the form of free text powernotes rather than structured data.

1. Pendlebury ST, Lovett N, Smith S, Dutta N, Bendon C, Lloyd-Lavery A, et al. Observational, longitudinal study of delirium in consecutive unselected acute medical admissions: age-specific rates and associated factors, mortality and re-admission. *BMJ Open* 2015;5:e007808-e007808.

2. Pendlebury ST, Klaus SP, Mather M, de Brito M, Wharton RM. Routine cognitive screening in older patients admitted to acute medicine: abbreviated mental test score (AMTS) and subjective memory complaint versus Montreal Cognitive Assessment and IQCODE. *Age Ageing* 2015;44:1000-1005.”

3. Study procedures: What are “EPR powerforms”? “Powernote entries”? Presumably something related to the Cerner system, but please give a brief description of what this means. Is the intention to make free-text entries available also?

Response: We have modified the text to clarify this and added text to the Supplement as follows:

Study procedures, page 8: “The database contains data from EPR power forms (i.e., electronic proformas from which structured data can be extracted) but not powernotes (i.e., free-text entries by clinicians, **see Supplement**).”

We have noted that free-text entries are currently unavailable as a limitation:

Discussion, limitations, page 17: “Finally, free text entries including the medical clerking and entries from allied health and specialist personnel are currently unavailable, although AI methods could eventually be used to extract analysable data as is done for mental health (CRIS) data [27].”

We have also added further explanatory text to the Supplement as follows:

“EPR Powerforms

EPR powerforms are bespoke electronic proformas “built” by the OUHFT EPR team and designed to capture structured clinical data. The cognitive screening form is an example of a powerform (see

supplemental figure). The data entered into such a powerform can be easily extracted and analysed including all the individual powerform items (eg the individual AMTS questions). The structured format of powerforms also allows the information contained within to be displayed automatically in different locations in EPR. For example, the cognitive screening powerform results populate the observations tab and the “cognitive assessments” tab where all the cognitive screening results over time and across different encounters are shown allowing clinicians to see the cognitive trajectory. This tab also displays the clinical frailty score assuming the relevant powerform has been completed, so that the cognitive assessments are put in the context of a global frailty assessment. The structured powerform data can also be included in algorithms for example, the AMTS score or untestability is included in the delirium susceptibility score algorithm which is calculated automatically within EPR and displayed in real time.³

EPR Powernotes

EPR powernotes are created by clinicians as electronic free text entries and are therefore entirely unstructured. Such free text data cannot be easily extracted or analysed and require the use of natural language processing algorithms. Information entered into a free text powernote, for example the AMTS score or documentation of delirium diagnosis, cannot therefore be automatically displayed in other parts of EPR or used in algorithms and quickly becomes “lost” amongst the mass of other entries.”

3. Pendlebury ST, Lovett NG, Smith SC, Wharton R, Rothwell PM. Delirium risk stratification in consecutive unselected admissions to acute medicine: validation of a susceptibility score based on factors identified externally in pooled data for use at entry to the acute care pathway. *Age Ageing* 2017;46:226-231.”

4. If date of birth is removed from the EPR information at the anonymisation stage, how is patient age calculated / represented in the database?

Response: Patient age at admission/assessment is included in the information provided by the OUHFT Information Analysts (Table 2).

5. The database starts from 65+ but the cognitive screening information starts aged 70+ - are there plans to expand the coverage down to 65+? Otherwise will be difficult to use patients aged 65-69 in many of the analyses which require cognitive information as a covariate? Or will alternative methods/data be used for these patients? Does this need to go into the limitations also?

Response. This is an important point. The age cut-off of 70+ years was chosen initially taking into account the number of cognitive screens required (given that screening would be required at scale) and the age-specific prevalence of cognitive frailty in acute hospital admissions (*BMJ Open* 2015;5(11):e007808-e007808). We also considered the guidance at the time which was for

dementia screening in those aged ≥ 75 years (dementia CQUIN/NHSE England FAIR process) and delirium screening in those aged ≥ 65 years (BGS/RCP). However, with the successful implementation of screening at scale and the recent GIRFT guidance, we have now reduced (from 2024) the screening age to 65 years and older so we shall have data on all those aged 65-69 going forwards.

We have added text as follows:

Discussion, limitations, page 17: "Third, patients aged < 70 years did not get routine cognitive screening and cognitive data are therefore limited to those deemed as at-risk by clinicians. However, the age cut-off has now been lowered to 65 years and over (2024 onwards) so cognitive data will be routinely available on those aged 65-69 years going forwards."

6. Outcome measures: Could this please be re-organised with the definition/detail of each outcome next to the bullet point for easier reading?

Response: We have re-organised the section as requested (Page 11).

7. The sentence "All participants are followed until death or the end of the specified study period" seems more appropriate for specific cohort studies which may be run using the data from the ORCHARD-EPR rather than a protocol for a research database, as presumably this is an open-ended database?

Response: We agree with the reviewer and have removed this sentence.

Typos:

Exclusion: "at-risk" change to "at risk"

Study size – should be "in 64,641 unique patients"? (remove 'over')

Response: We thank the reviewer for pointing these out and we have corrected these typos.

Reviewer: 2

Dr. Kate Best, University of Leeds Faculty of Medicine and Health

Comments to the Author:

The protocol describes an very interesting data source that includes variables not available in other secondary care data sources such as HES.

We thank the reviewer for their positive comments.

I have a few comments to improve clarity:

1. Please could you add a basic description of powerform and powernote.

Response: We have added further text including to the Supplement. Please see our response to Comment #3 for Reviewer 1 above.

2. Estimation of prevalence is listed as a key objective- could you clarify that this refers to hospital prevalence.

Response: We agree and have revised the text as follows:

Aims and Objectives, page 5: "1. Estimate the hospital-wide prevalence of cognitive frailty (i.e. delirium, dementia, delirium superimposed on dementia, low cognitive test score) by age, sex, and specialty."

3. ORCHARD-EPR is described as a data resource for researchers- are there plans to make the data accessible to those outside of the research team?

Response: We are currently looking into whether it is possible to modify the ethics approvals to widen access to include researchers from outside the University of Oxford and the OUHFT. We have added the following text:

Methods and Analysis, Regulatory approvals page 7: "Currently, the ethics approval for this study does not allow access to data to researchers outside the University of Oxford/OUHFT but widening access to researchers from other institutions is currently being explored."

4. Are there plans to record ethnicity?

Response: Yes, this is recorded in EPR and therefore it should be possible to include this information in the database going forwards. We have now included ethnicity in Table 2 under demographics. We would plan to do some further work to determine the accuracy of these data against manually acquired audit data as the accuracy is currently unclear.

VERSION 2 – REVIEW

REVIEWER	Fogg, Carole University of Southampton, School of Health Sciences
REVIEW RETURNED	29-Apr-2024
GENERAL COMMENTS	Many thanks for the responses and edits.
REVIEWER	Best, Kate University of Leeds Faculty of Medicine and Health, Academic Unit of Ageing and Stroke Research
REVIEW RETURNED	26-Apr-2024

GENERAL COMMENTS	Thank you for the detailed responses and changes based on the reviewers' comments. Congratulations on a great protocol.
---